# Laser Fabrication of Anti-Icing Surfaces: A Review

**DOI:** 10.3390/ma13245692

**Published:** 2020-12-13

**Authors:** Annalisa Volpe, Caterina Gaudiuso, Antonio Ancona

**Affiliations:** 1Department of Physics, University of Bari “Aldo Moro”, 70125 Bari, Italy; 2National Research Council-Institute for Photonics and Nanotechnologies, Via Amendola 173, 70125 Bari, Italy; caterina.gaudiuso@uniba.it (C.G.); antonio.ancona@uniba.it (A.A.)

**Keywords:** anti-icing, laser, microstructuring, wettability, superhydrophobicity, metals, polymers, LIPSS, DLIP, direct laser writing

## Abstract

In numerous fields such as aerospace, the environment, and energy supply, ice generation and accretion represent a severe issue. For this reason, numerous methods have been developed for ice formation to be delayed and/or to inhibit ice adhesion to the substrates. Among them, laser micro/nanostructuring of surfaces aiming to obtain superhydrophobic behavior has been taken as a starting point for engineering substrates with anti-icing properties. In this review article, the key concept of surface wettability and its relationship with anti-icing is discussed. Furthermore, a comprehensive overview of the laser strategies to obtain superhydrophobic surfaces with anti-icing behavior is provided, from direct laser writing (DLW) to laser-induced periodic surface structuring (LIPSS), and direct laser interference patterning (DLIP). Micro-/nano-texturing of several materials is reviewed, from aluminum alloys to polymeric substrates.

## 1. Introduction

Ice in its several forms, i.e., frost, glaze, rime, snow, can cause severe problems for locks and dams [1], solar panels [2], wind turbines [3], aircraft [4,5], heat pumps [6], power lines or telecommunication equipment [7], civil engineering materials [8], and oil platforms [9], especially when it adheres and accumulates. These problems can increase energy consumption, reduce the energy conversion efficiency, origin mechanical and/or electrical malfunctions, and define safety hazards [10,11]. Among all the mentioned fields, the aerospace industry is particularly affected by icing phenomena, which can occur both while aircraft are on the ground [12] and in the air [13]. In the first case, before flying, ice needs to be removed, since it alters aerodynamic properties, and ice fragments can be sheared off by aerodynamic drag and fly into the engines. In the latter case, the impact of supercooled water droplets found in clouds, depending on atmospheric conditions, may stick to the surface, creating a dangerous ice layer. Therefore, the overall recommendation of the safety authorities is to strengthen the ice protection systems (IPS) to all commercial aircraft [14]. These protection systems can be classified into two groups: anti-icing systems, which aim to prevent ice formation, and de-icing systems, which aim at the removal of already-accumulated ice or frost from a surface. 

In recent years, great attention has been paid to the anti-icing surfaces, as attested by the increasing number of research papers and patents related to the topic “anti-icing” (Figure 1).

It can be difficult to find a common definition of anti-icing, but some ubiquitous features must be exhibited by such surfaces, i.e., (i) inhibition of water condensation on the surface; (ii) inhibition of the incoming water’s freezing; and (iii) weakening of the adhesion strength of ice, when formed, so that it can be easily removed [16]. 

Different strategies have been employed to prevent ice accretion and to easily remove it, and they can be divided into active and passive approaches. 

Keeping the surface temperature above the freezing point by means of electro-thermal heating is an efficient active method to reduce the likelihood of ice formation or to enhance its melting [17]. The exploitation of the Joule effect for heating line conductors is at present recognized as the most efficient engineering approach [18], especially in the case of transmission lines. However, this strategy is extremely time and energy consuming. In addition, electromagnetic disturbance can be generated by the passage of electric flow, which could interfere with the operation of the apparatus. Another method is to use freezing point depressants. It is well known that salt depresses the freezing point to the eutectic point, facilitating the ice in melting. Instead, in order to prevent icing and frosting of water on aircrafts surfaces, organic liquids are exploited, thanks to their lower crystallization temperatures compared to plain water [13]. In particular, for commercial aircraft in the northern hemisphere, various de-icing fluids are used [12,19]. They are essential not just on the exposed surfaces of aircraft wings, but also on other aerodynamic areas such as under the wings and on the rear spar stabilizer areas. Two standard test methods, namely Wet Spray Endurance Test [20] and Boundary Layer Displacement Thickness [21] are employed in climatically controlled wind tunnels to give a measure of the remaining film thickness on a surface during its continuous removal under an increasing flow of air over it. Both tests are critical for safe flight operation and are closely associated with density, surface tension and viscosity of the exploited fluids. 

This notwithstanding, the use of chemical agents to prevent ice formation has many drawbacks. First, their effectiveness is short-lived. Therefore, for large surfaces, periodic treatments are required. Furthermore, extensive use of these liquids, besides having high costs, can cause environmental problems [22].

Other active de-icing techniques consist of using mechanical forces to remove ice accretion from surfaces. With this aim, direct scraping or mechanical removal by shock waves, vibrations, or the twisting of conductors are applied. However, these methods often require personnel to reach the lines and towers; furthermore, helicopters or even shotguns must also be exploited when the ice is less accessible [18]. Moreover, applying mechanical force during de-icing causes additional stress to the networks, which in some cases can lead to failure. Consequently, such methods are certainly neither safe nor efficient. In addition, many other anti-icing and deicing methods have been studied and applied, e.g., electro-impulse systems, shape memory alloys, and ultrasound technology [23].

In addition, hybrid solutions consisting of material with a combination of passive anti-icing and active deicing functionalities have also been reported [24]. In [25], the prevention of freezing above −14 °C without any source of power input was claimed in the case of incoming water by using a spray-coating of perfluorododecylated graphene nanoribbons (FDO-GNRs), whereas resistive heating represents an alternative active deicing method in more extreme sub-zero environmental conditions. 

Thus, most of the active or hybrid strategies striving against ice formation are apparently inefficient, energy-consuming, expensive, or dangerous to the environment. It is thus crucial to develop innovative, green, cost-effective, and efficient technologies for anti-icing and deicing [11]. 

A new development perspective for pure passive anti-icing methods is surface micro- and nano-structuring. Recently, several efforts have been devoted towards the modification of surface topography and/or chemistry in order to obtain superhydrophobic properties [26,27]. A close relationship between the water repellency of superhydrophobic surfaces (SHSs) and icephobicity has been demonstrated [28]. Indeed, the air trapped between the water droplets and the underlying surface texture allows the surfaces to exhibit not only high water contact angle (WCA) and low contact angle (CA) hysteresis [29], but also prevent ice formation because of reduced water adhesion. Consequently, using SHSs for anti-icing application is undoubtedly an inherently attractive strategy. Among the different technologies available for surface micro- and nano-structuring, short and ultrashort laser pulses have several advantages: (i) they offer extreme flexibility in the morphology of the structures and surface micro-/nano-features that can be created [30,31,32,33]; (ii) in principle, they have no limitations in terms of manufacturable materials, as long as the targets absorb the laser wavelength [34,35]; and (iii) it is possible to scale up such technology [36] to structure large areas with industrially relevant process times and costs [37].

A comprehensive review of the most relevant works on the use of laser technology to functionalize surfaces with anti-icing properties is still lacking and would be beneficial to assess the latest advances and evaluate how mature is such a technology to be employed at an industrial scale.

In the first part of this review, the key concepts regarding wettability, superhydrophobicity and anti-icing properties will be highlighted. Then, the main laser techniques used to fabricate anti-ice surfaces will be illustrated, ranging from Direct Laser Writing (DLW) to Laser-Induced Periodic Surface Structuring (LIPSS) and Direct Laser Interference Patterning (DLIP). The principal laser treatments carried out to obtain superhydrophobic and anti-icing substrates will be discussed in depth either for metallic or polymeric substrates. 

## 2. Surface Wettability

### Key Concepts

Wettability is the result of the stability of the three phases, i.e., gas, liquid and solid, which come into play when a liquid lies on a solid surface. In fact, while the force at the solid–liquid interface (adhesive) tends to spread the liquid, that within the liquid itself (cohesive) instead resists spreading. Therefore, the balance between such forces determines the ability of the surface to be wet. The wetting behavior is normally evaluated by considering the contact angle (CA), i.e., the angle θ between a droplet deposited on the surface and the surface itself, as shown in Figure 2. 

In the early nineteenth century, Young found a relationship relating the intrinsic contact angle θ to the surface tensions of the liquid–gas $\gamma_{lg}$, solid–gas $\gamma_{sg}$, and solid–liquid interfaces $\gamma_{sl}$, which can be expressed as [38]:(1)$$\gamma_{sl}=\gamma_{sg}-\gamma_{lg}cos\theta$$

This relation, also called the Young equation, is the starting point for all the investigation on the wetting behavior of surfaces.

As mentioned before, another important concept to be introduced to investigate the wettability is the adhesion of the water droplet to the substrate, addressable through the work of adhesion $W_{a}\;$ [39]. This can be defined as the work done when two surfaces are separated to a distance of infinity and, in case of liquid–solid combinations, can be expressed as:(2)$$W_{a}=\gamma_{sg}+\gamma_{lg}-\gamma_{sl}$$

From (1) and (2), $W_{a}\;$ can be also expressed as:(3)$$W_{a}=\gamma_{lg}\left(1+cos\theta \right)$$

A low contact angle (θ<90°) indicates hydrophilicity, namely good wetting. If the angle approaches zero, $W_{a}=2\gamma_{lg}$, and total wetting is observed with the formation of a liquid film. When θ>90°, hydrophobic properties are revealed, while for θ>150°, the surface is classed as superhydrophobic. No wetting and no adhesion are instead characteristics of a surface where  θ=180°, that is, $W_{a}$ becomes zero.

For non-ideal surfaces, where non-homogenous chemical composition and/or non-uniform topography are present, wettability cannot be described by Equation (1). Consequently, the apparent contact angle $\theta^{*}$ [40] is considered instead of θ. Considering real solid rough surface, two possible configurations can be sketched for a deposited droplet: (i) it adapts to the contours of the real solid surface, conforming to the peaks and valleys of its roughness (Wenzel state, Figure 3a); or (ii) it stands on the nails of the roughness, leaving only air to fill the cavities below (Figure 3b). This latter condition, the Cassie and Baxter state, is also called the “fakir” state, since the droplet lies on micro asperities [41].

In the Wenzel state, the apparent CA depends on the Young’s CA through the Wenzel wetting equation [42]:(4)$$cos\theta^{*}=r\;cos\theta$$
where *r* is ≥1, and it is defined as [42]:(5)$$r=\mathrm{rouhgness}\;\mathrm{factor}=\frac{\mathrm{actual}\;\mathrm{surface}}{\mathrm{geometrical}\;\mathrm{surface}}$$
namely, the ratio of the effective solid surface area to the apparent projection area of a rough surface [43].

The Wenzel Equation (4), perfectly suitable for characterizing hydrophilic and hydrophobic surfaces, reveals that the contact angle $\theta^{*}$ of a real surface can be increased by increasing its roughness. Nevertheless, this model fails to describe superhydrophobicity, because it does not consider the air pockets that are bounded between the surface and liquid. Consequently, another model was introduced to explain many wetting phenomena, such as the lotus effect [44] and self-cleaning behavior [45], obtainable when CA>150°, developed by Cassie and Baxter.

In case of superhydrophobicity, the droplet no longer infiltrates the surface roughness, but a kind of combined phase contact is established between the solid–liquid and gas–liquid interfaces. So they introduced a new relationship for the apparent contact angle [46]:(6)$$cos\theta^{*}=f_{1}cos\theta_{1}+f_{2}cos\theta_{2}$$
where $\theta_{1}$ and $\theta_{2}$ are, respectively, the Young contact angles of the liquid droplet on the solid and gas bearings, while $f_{1}$ and $f_{2}$ are the percentage of the solid–liquid and gas–liquid contact area, with $f_{1}+f_{2}=1$. Therefore, if $\theta_{2}=180{}^{\circ}$, Equation (6) becomes:(7)$$cos\theta^{*}=f_{1}\;cos\theta_{1}+f_{1}-1$$
which explains some non-wetting superhydrophobic phenomena [47]. The existence of a critical value of roughness factor was found to dictate the transition between the two mentioned wetting states, as found Dettre and Johnson, who stated that beyond it, wettability is better described by the Cassie–Baxter model than by Wenzel [48].

Besides the apparent contact angle $\theta^{*},$ contact angle hysteresis (CAH) gives valuable information on the moving properties of a droplet on a solid surface [49]. When a surface is inclined, an advancing and a receding front, forming different angles with the surface, can be recognized as the droplet slides. The difference  Δθ between these two angles, the advancing $\theta_{A}$ and the receding $\theta_{R}$ (Figure 4), defines the CAH. When Δθ<10°, the droplet can be easily removed from the surface; conversely, when Δθ is sufficiently large, the liquid viscosity is such that the droplet cannot readily roll off the surface. 

Another angle that characterizes a moving droplet is the rolling angle α (see Figure 4), also called roll-off angle or sliding angle SA, namely, the minimum angle at which a liquid droplet would begin to slide down an inclined surface. It is related to several factors, such as CA, CAH and droplet size [50,51], and can be described by the following equation, where the sliding angle is defined depending on the advancing and receding contact angles ($\theta_{a}$ and $\theta_{r}$, respectively) [52]:(8)$$\;sin\alpha =\frac{\gamma Rk}{mg}(cos\theta_{r}-cos\theta_{a})$$
where *γ* is the surface tension, R and k are length scale and shape constants for describing the droplet profile (R is generally taken to be the drop radius, while k is an empirical parameter obtained by fitting the experimental data), respectively, *m* is the mass of the droplet, and *g* is the gravitational acceleration. Thus, a small hysteresis corresponds to a small SA, and researchers usually correlate water adhesion to CHA or SA interchangeably.

Usually, the total wetting Wenzel regime characterizes sticky surfaces with high CAH, where the water droplets tend to grip more than to a flat equivalent surface. Conversely, surfaces which behave in the Cassie and Baxter state are slippery and are characterized by a low CAH, which leads the water drop to slide away more easily than an equivalent flat surface. However, the existence of “sticky” surfaces, i.e., with high intrinsic contact angle hysteresis, in the regime of Cassie and Baxter has also been reported [53,54].

## 3. Anti-Icing Properties

In recent years, scientists have found that a key factor to obtaining anti-icing properties is superhydrophobicity, which can be obtained through low surface energy coatings [55] or by increasing the surface roughness [56,57,58]. Dwelling on the latter strategy, the bio-inspired example of SHS, such as surfaces reproducing the doubled micro-nanostructures of the lotus leaf, have attracted great attention [59,60,61,62,63]. 

Tourkine et al. showed that using SHS at subzero temperature made it possible to easily remove a deposited water drop before freezing thanks solely to gravity, thus preventing the formation of an ice layer [64]. Furthermore, Kulinich et al. [29] suggested that only when the CAH was low would the correspondence between low ice adhesion and high CA be consistent, suggesting icephobicity to be related to the CAH rather than to superhydrophobicity alone. Indeed, higher CAH indicates higher water–solid adhesion and, consequently, at subzero temperature, when the water becomes ice, higher ice adhesion, too. On the contrary, water-repellent surfaces, i.e., those that exhibit high water CAs and low CAHs, remain dry because of the easy water roll off, and thus, at subzero temperature, ice-formation is reduced. 

The main aspects investigated to assess the anti-icing properties of a surface are:icing delay, intended both as a time delay in the solidification of the water and as a reduction in the threshold temperature for freezing. A lower threshold for the formation of ice would, indeed, allow wider range of operating temperature conditions. Similarly, a postponement of freezing would increase the probability of water removal before solidification.interaction between water and the solid surface. That is, if before freezing occurs, water droplets were to roll off the surface, there would be no chance of ice being formed. This phenomenon is strictly correlated with the sliding angle.ice adhesion. Although ice is formed, its adhesion should be minimized, so that a small external force or, e.g., in case of a wind turbine, the centrifugal force of its own blades, is sufficient to remove the frozen layer, facilitating the de-icing process.

Usually, the testing of these three aspects of an anti-icing surface takes place at atmospheric pressure and in static conditions. Consequently, in the aerospace field, these tests are more appropriate for verifying the anti-icing formation on the ground than in the air, where the atmospheric conditions are more extreme. 

### 3.1. Icing Delay Time

Several authors have tested the anti-icing propension of processed surfaces by evaluating the icing delay time (DT), namely, the time needed for a reference drop to freeze [65,66]. The procedure they follow is simple: a reference droplet is placed on a surface that has been precooled (generally in a range from −20 to −5°C), or, alternatively, the surface is cooled after the droplet deposition. Initially, the reference droplet is transparent, and after a certain DT, it becomes non-transparent, indicating transition to the frozen state. Usually, SHSs have a longer DT compared to reference surfaces, and thus better anti-icing properties [67]. 

A suitable explanation for such phenomena and of the anti-icing properties of SHSs when the temperature is kept below 0 °C is provided by the classic nucleation theory, together with the heat transmission between superhydrophobic surfaces and water droplets [68,69]. In fact, upon a cold surface at a temperature *T*, heterogeneous nucleation occurs at the liquid–solid interface and, according to the classical nucleation theory [70], the nucleation rate *R* can be expressed as [71]:(9)$$\log_{10}R\left(T\right)=\log_{10}J_{0}-\frac{\Delta G}{2.303T{\left(T_{m}-T\right)}^{2}},$$
where
(10)$$\Delta G=\left(\frac{2\pi r^{3}}{3}\Delta g+2\pi r^{2}\gamma_{sl}\right)(2-3cos\theta^{*}+\cos^{3}\theta^{*}),$$
is the nucleation free-energy barrier for the heterogeneous nucleation, $J_{0}$ is a pre-factor, $T_{m}=273.15\;K$ is the temperature at the water freezing point, *r* is the droplet radius, $\gamma_{sl}$ is the solid–liquid interfacial tension, Δg is the difference between Gibb’s energy density of ice and liquid water, and $\theta^{*}$ is the apparent contact angle, as introduced in Equation (7). From Equations (9) and (10), it emerges that the lower the temperature and the free-energy barrier, the faster the nucleation rate. Moreover, it can be seen from Equation (10) that, being proportional to $\left(2-3cos\theta^{*}+\cos^{3}\theta^{*}\right)$, ΔG increases with $\theta^{*}$, leading to a smaller nucleation rate and, consequently, deferring the ice formation process. Hence, superhydrophobic surfaces delay the solidification of condensed droplets compared with hydrophobic ones. 

Moreover, it should also be considered that for SHS in the Cassie state, being a good thermal insulator, the air trapped between the water droplet and the rough surface contributes to a reduction in the heat transfer between the liquid surface and solid [72], further retarding the icing process. The solid–liquid–air three-phase interfaces need to be taken into account for describing, from a thermodynamic point of view, the phenomena occurring when a droplet is suspended over the surface. Heat conduction causes heat flux between the droplet and the cold surface, while thermal radiation leads to heat release between the droplet and air [73]. 

The net heat gain in unit time and volume Δh can be expressed as [65]:(11)$$\Delta h=h_{g}+h_{g^{\prime }}-h_{l}-h_{l^{\prime }}\;,$$
where $h_{g}$ and $h_{g^{\prime }}$ are the heat gain terms due to heat conduction and thermal radiation, respectively. Analogously $h_{l}$ and $h_{l^{\prime }}$ are the heat losses per unit time and volume due to heat transfer and thermal radiation. In general, the heat loss per unit time through heat conduction is greater than thermal radiation, which can be neglected [65]. The time required for a water droplet to completely freeze $t_{f}$ is related to the net heat gain Δ*h* by the following equation [74]:(12)$$t_{f}=\rho \frac{C_{p}\left(T_{0}-T_{s}\right)}{\Delta h}\;\;,$$
where ρ is the water density, $C_{p}$ the heat capacity, $T_{0}$ the initial temperature of the droplet and $T_{s}$ the surface temperature. When a droplet is suspended upon a rough processed surface in a Cassie–Baxter state, the contact area between the liquid and the solid is smaller than on the analogous ideal surface. Therefore, a smaller amount of heat is lost through heat conduction to the sample surface. Likewise, a larger liquid–air contact area is obtained, which enhances the heat released from air. For the above-mentioned reasons, Δh on the rough surface is larger than that of a reference flat surface causing a slower freezing, as emerged from Equation (12).

### 3.2. Short Contact Time Interation

Minimizing the contact time between a water droplet and a surface at temperatures below the freezing point has been reported to be a fruitful method for preventing ice accretion from incoming water [75]. In fact, in this case, some naturally occurring icing events are induced by the impact of supercooled water droplets onto surfaces, and are commonly referred to as “freezing rain”, “atmospheric icing”, or “impact ice” [67]. 

A superhydrophobic surface, characterized by a low sliding angle and low CAH, which would make the overcooled water droplets bounce or roll off, would behave like a good icephobic surface [57]. 

Bahadur et al. [76] developed a model to describe in detail the ice formation in the case of a droplet impacting a structured SHS. The model incorporates the droplet contact time, heat transfer and heterogeneous nucleation theory. According to the model, when a droplet comes into contact with a supercooled surface, the surface micro-asperities act as nucleation sites for ice crystals. This causes a decrease in the retraction force connected to the apparent receding contact angle that, eventually, leads to the incomplete retraction, pinning, and freezing of the droplet. However, ice formation is inhibited whenever the droplet contact time is shorter than the time necessary to induce pinning. This transient model was found to consistently describe the experimental results. More generally, in order to predict the behavior of a surface in terms of its resistance to ice formation, the model demonstrated that multiple dynamic processes must be integrated, such as dynamic wetting, heat transfer and nucleation theory.

In Figure 5 [77], a typical setup used to study ice accumulation under dynamic water flow conditions is shown. Water droplets (usually kept at temperatures close to 0 °C) are dropped through a capillary onto a sample surface that is tilted at a certain angle by a remote controllable micromanipulator stage. The distance between the water droplets injection system and the sample substrate is accurately handled and is usually less than 10 cm. The target surface is kept at a controlled temperature (between −35 °C and room temperature), either by a thermoelectric cooling stage [57] or by a climatic chamber [77]. The droplet temperature is supposed not to change significantly during the short time-of-flight. The dry environment in the chamber (<5% humidity) and low condensation on the investigated substrates are sometimes ensured by an air flow. A tailored water flow through the capillary is enabled by a pump. An optical imaging system, preferably a high-speed video camera, is used to follow the motion of the droplets hitting the substrates and ice deposit left behind. 

### 3.3. Ice Ahdesion on a Surface

Once ice forms on a surface, its interaction with the substrate at the atomic or molecular scale is a combination of short-range interaction forces, namely electrostatic forces, hydrogen bonding and van der Waals forces. In addition, ice infiltration and interlocking with microscopic surface asperities concur to mechanical adhesion [67,78]. 

The ice adhesion to the substrate can be measured by different procedures depicted in Figure 6.

In the Peak force method [81] (Figure 6a), a test column is placed on the icephobic substrate. The water poured into the column freezes and adheres to the substrate. A force meter measures the required force *F* to detach the ice column from the icephobic substrate. Here, ice adhesion strength is determined by dividing the maximum breaking force value by the cross-sectional area of the ice–substrate interface.

In the centrifugal force method depicted in Figure 6b, a tip of a wing or a beam is coated with icephobic material [82,83,84], and the formation of ice is induced through a rain of sub-cooled droplets or through a cuvette ice column. By gradually increasing the rotational speed of the beam, a shear force is induced at the interface between the ice and the ice-phobic substrate. Ice adhesion strength is thus determined as the centrifugal force when detachment from the substrate occurs along with cross-sectional area of the detached ice. 

In the tensile force method [85] shown in Figure 6c, the gap between two concentric aluminum cylinders is filled with water. An ice-phobic material coats the inner surface of the large cylinder. Water solidification is induced by keeping the whole apparatus in a freezer at a specified sub-zero temperature. Once ice forms between cylinders, a tensile machine allows to apply a pulling force to the inner cylinder. In this case, the ice adhesion force is obtained as the ratio between the pulling force at the detachment point divided by the area of ice-icephobic interface.

Another method to test the ice adhesion based upon the principle of harmonic excitation was developed by Strobl et al. [80] (Figure 6d). Through the use of a dynamic vibration measurement technique with an electromagnetic shaker, the interfacial forces between ice and the surface can be characterized by means of a strain gauge, which detects the failure as a change in the bending stiffness of the composite beam.

The methods reported so far to assess the anti-icing properties are the main ones used by researchers to characterize the anti-icing surfaces fabricated according to the methods described in the next section.

## 4. Fabrication of Superhydrophobic Surfaces with Anti-Icing Properties

### 4.1. Non-Laser Methods

The anti-icing properties of surfaces depend on their chemical composition and roughness [41]. Both characteristics can be modified in several ways in order to obtain a surface with passive anti-icing properties [41,86]. 

So far, the role of surface roughness has been highlighted, but it is worth noting that non-rough substrates with a low surface energy, such as PTFE (or Teflon^®^) and PDMS, seem to be more prone to low ice adhesion [78]. Chemical methods have been developed to confer materials with more suitable surface chemistry to induce water repellency and reduce ice adhesion. The explored chemical methods mainly consist of the deposition of low surface energy polymer coatings and can be classified into [86]: (i) self-assembled monolayers with –CH3 or –CF3 groups oriented outward to the surface reducing the access of water droplets to the substrate; (ii) coatings disrupting the structure of a liquid-like layer thanks to a heterogeneous chemical composition of at least two highly hydrophobic components [66,87]; and (iii) porous or superhydrophobic deposits that stimulate the formation of minuscule air pockets that, in icing conditions, inhibit adhesion by creating stress concentrations [78]. 

Superhydrophobic coatings with anti-icing properties have also been demonstrated using suspension plasma spraying (SPS) of titanium dioxide [88]. Icing wind tunnel experiments showed that, compared to commercial superhydrophobic coatings, the SPS coatings exhibit substantially superior performance in icing/deicing cyclic tests.

However, it has been found that, even conveniently designing the chemistry of the smooth surfaces, it is hard to achieve ice adhesion regimes sufficiently low to be of practical interest [89]. Therefore, increasing the surface roughness is an essential prerequisite for obtaining icephobic surfaces. In this case, increasing the percentage of trapped air between the liquid–solid interface reduces the contact area and the heat loss through heat conduction, as explained in Section 3.1. Micro-/nanostructures with controlled topographies that increase the trapped air can be fabricated by electrochemical etching [90], photolithography, or other analogous mold-based lithographic techniques [91]. Wang et al. [92] integrated soft-lithography and hydrothermal methods to obtain hierarchical micro-/nanostructures composed of poly(dimethylsiloxane) (PDMS) and zinc oxide (ZnO). These surfaces show robust superhydrophobicity and can be easily de-iced.

A reduction up to 50% of the ice adhesion has been also demonstrated exploiting a photolithographic process that introduces sub-structures into smooth polydimethylsiloxane coatings, which stimulates the initiation of macro-cracks [93]. Others groups have exploited the method of ZnO nano-hair planting [73], combined with machine processing. In this way, the authors conferred icephobic properties to stainless steel surfaces.

A series of micro-cubic arrays were fabricated by Hou et al. [94] on silicon surfaces by selective plasma etching. The effect of surface microstructure size on anti-icing and icephobic performance were explored in terms of icing delay time and ice adhesion strength. Thanks to the air pockets entrapped in the microstructures, a time delay in the ice formation two orders of magnitude greater than that measured on the pristine surface was achieved, together with a reduction of the ice adhesion strength.

However, all these non-laser-based techniques suffer from intrinsic weaknesses, such as complexity, toxicity of the chemical reagent, and/or expensive mask fabrication requiring expensive clean rooms. To overcome these drawbacks, many groups are moving to the laser machining of surfaces to confer them with anti-icing properties.

### 4.2. Laser Micro-/Nano-Structuring of Surfaces

Laser machining is a one-step direct fabrication method that makes it possible to produce features on a micro and/or nano scale on a broad range of substrates such as metals [54,95], polymers [96,97] and glasses [98]. Laser-based techniques have been demonstrated to be an outstanding method for obtaining surfaces with controllable superhydrophobic and anti-icing properties [99] without the environmental impact of chemical methods, and with high flexibility, without the need for expensive masks or molds. 

Based on the dimensions of the structures to be obtained, several laser techniques can be exploited for surface modification. Starting from the micro scale to the nano scale, Direct Laser Writing (DLW, features from a few hundreds of micrometers to about 10 μm), Direct Laser Interference Patterning (DLIP, features from about 10 μm to 1 μm), and Laser-Induced Periodic Surface Structuring (LIPSS, features <1 μm) are the main laser techniques described in literature. Hybrid techniques that combine two of the previous three to get hierarchical structures at the micro/nano scale or that mix laser and chemical modification of the substrate have also been reported [100].

Starting from the micro scale, direct laser writing is a versatile tool for modifying the surface topography of a solid substrate [101]. Laser surface structuring can be achieved either by fast scanning the beam over large areas using galvanometric mirrors and F-Theta lenses, or by moving the samples below the focusing optics. The choice of the irradiation parameters, such as pulse duration, laser wavelength, pulse energy, repetition rate, beam polarization and diameter, determines the type of modification achieved. Other important process parameters that determine the dose of energy released onto the sample, and thus its modification, are: the scan velocity *v* and the focused spot diameter, which, together with the pulse repetition rate, determine the pulse-to-pulse spatial overlap, the number of overlapped scans (OS), and the hatch distance h (i.e., the lateral spacing between consecutive parallel scanned line). 

Direct laser ablation for reproducing specific topographies on surfaces has been demonstrated by several authors to be a quick and flexible way to obtain the superhydrophobicity of the lotus leaf. Several laser sources have been employed, from nanosecond [102] to ultrashort laser pulses [103]. Short pulses represent a cheaper choice with respect to ultrashort ones, which are preferable in cases where high precision and low thermal effect are required [104]. Besides the pulse duration, great attention has been devoted to finding the right laser parameters and geometry, e.g., changing the speed, the number of scanning loops or the hatch distance.

Several studies have been concerned with the use of DLW combined with chemical treatments for enhancing the superhydrophobic [100,105] and anti-icing properties of surfaces [106]. A single-step direct laser texturing process, without any chemical modification, remains a challenging task, and most of the works are limited to studying the superhydrophobic properties of the laser-treated materials [107]. 

Among all the materials, much attention has been directed towards the use of DLW to confer anti-icing properties on metal surfaces. In particular, considering their light weight, aluminum alloys are used extensively in aircraft production, especially for the wings and fuselage, the safety of which is greatly threatened by ice formation [108]. A laser texture pattern that, despite its simplicity, has been demonstrated to be effective against ice accretion consists of a series of parallel scanned lines, as shown by Xing et al. [65]. They used a picosecond laser source emitting a Gaussian profile beam at a wavelength of 1030 nm and a repetition rate of 500 kHz. They used an output laser power of 20 W and scanned the beam at a speed of 3 m/s. Using a hatch strategy in the horizontal direction, with a lateral pitch distance h ranging from 20 to 100 μm, micro-nano tertiary hierarchical structures consisting of micro-grating (Figure 7a), cauliflower-like protrusions and nano-structures (Figure 7a-inset) were obtained. All the laser-machined samples exhibited CA above 160°, showing a superhydrophobic behavior compared to the original aluminum alloy surface whose CA was about 106.1°. Moreover, all the superhydrophobic surfaces showed SAs below 10.0°. In their work, Xing et al. [65] also tested the anti-icing behavior of the laser processed surfaces that exhibited the best water repellency. In particular, the freezing time delay DT of the water droplets deposited on the superhydrophobic laser textured aluminum alloy surfaces was measured and compared to a flat reference surface. As shown in Figure 7b, with the decrease of the temperature of the surrounding environment down to −22 °C, the droplets on the laser-machined surfaces started to lose transparency after 4015 s and were fully frozen after 4253 s, which is a significant delay of 1147 s compared to the reference surface. In addition, it was noticed that, in the case of textured surfaces, the transition to the frozen state started at a temperature 6 °C lower than the original surface. The authors explained the above results on the basis of thermodynamics. According to the Cassie model, the droplet is lifted by the parallel micro-gratings and cauliflower-like protrusions. Here, the trapped air is crucial for the anti-icing properties, as highlighted in Section 3.2. In fact, the smaller liquid–solid contact area and higher percentage of trapped air of the laser machined surface reduce the heat loss through contact heat conduction, in comparison with the original sample. Furthermore, the smaller the liquid–solid contact area, the larger the liquid–air contact area, thus improving the heat obtained from the air. From the above analysis, it emerges that net heat obtained for the droplet in unit time (Δ*h*, Equation (11)) on the micro-nanostructured surface is greater than that on the original sample surface. Therefore, according to Equation (12), the increase of Δ*h* results in the decrease of *t_f_*, which essentially delays the water droplet freezing and promotes the anti-icing ability of the laser-textured aluminum alloy surface.

Wang et al. [109] reported on a one-step laser method to obtain superhydrophobicity and anti-icing properties on stainless steel surfaces. Regular micro-grid cells with different hatch distances (from 60 μm to 300 μm) were generated by ultrafast laser scanning in two perpendicular directions (Figure 8a). The laser ablation induced the generation of numerous nanosized particles with an average diameter smaller than 500 nm, which redeposit onto lateral untreated areas, accumulating into rough micro-aggregates (Figure 8a—insets). For all hatch distances, a robust superhydrophobicity with durable self-cleaning performance was found after exposure to ambient air for 24 h. These properties were attributed to the formation of the hierarchical micro-/nanostructures and to the presence of a nonpolar carbon layer on the surface [110]. The evolution of water droplets set on thus-prepared superhydrophobic surfaces was monitored at an ambient temperature of −8.5 ± 1 °C and a humidity of about 30% (Figure 8b). The water droplets on the structured surfaces retained ellipsoidal shapes in liquid state for more than 500 min. After this time, the droplets wholly evaporated. In fact, they benefited from the superhydrophobicity conferred by the laser treatment, according to the classic nucleation theory and the heat transfer between the rough solid surface and water droplets (see Section 3.1). For comparison, the water droplet set on the original surface was also monitored. Here, freezing was already reached after 70 min at subzero temperature.

A similar microgrid structure fabricated by fs-laser was investigated in [111]. Here, the anti-icing properties of laser textured Al2024 surfaces were studied in depth as a function of the microgrid spacing h. Below a hatch spacing of 200 μm, a superhydrophobic behavior was found for 10 μL droplets. These samples were also demonstrated to have a self-cleaning behavior. The presence of contaminant particles has been proved to accelerate ice nucleation [112]. Thus, the easy removal of impurities can be exploited in order to obtain a surface with improved anti-icing properties. The hydrophobic behavior of the textured surfaces was also proved at sub-zero temperatures. In particular, after an icing/de-icing cycle, the droplets set on the textured substrate recovered their original shape, thus avoiding the surface wetting. Conversely, on the pristine sample, the droplets spread over the surface. In this case, a further temperatures reduction below zero would create a wild and dangerous ice layer. Exploiting a setup similar to that in Figure 5, a robust dynamic anti-icing behavior was also found at −20 °C. As shown in Figure 9a, a droplet falling onto the untextured Al2024 adhered to and wet the surface due to the hydrophilic nature of the substrate, freezing after a certain time. On the contrary, the robust water-repellent behavior of the textured surfaces even at sub-zero temperature prevented the droplet from adhering to the surface, leaving the material dry (Figure 9b).

Besides metallic substrates, the anti-icing properties of polymeric materials treated by DLW are attracting a great deal of attention [113]. In particular, polytetrafluoroethylene (PTFE) is a low-surface-energy polymer with outstanding physical and chemical properties, e.g., resistance to alkali and acid, almost total insolubility in all solvents, low friction coefficient, resistance at high temperature, low dielectric constant, and excellent charge storage stability. PTFE is extensively employed in many fields, such as in aerospace, textiles, and electronics. In [114], the PTFE surface was processed by a CO_2_ laser. Micro channel patterns with various spacing from 100 to 500 μm were produced and different laser powers (up to 5 W) and overlapped scan steps were tested. It was reported that an increase in the micro channel distance caused a decrease in the contact angle in both directions (parallel and perpendicular to the channel direction). Hierarchical micro-structures were obtained, and the surface micro-roughness was appropriately maximized step by step. A self-made monitoring system (similarly to Figure 5) was exploited to investigate the dynamic anti-icing behavior of the laser-textured PTFE anisotropic superhydrophobic surfaces. For this test, the authors used the sample exhibiting the highest CA of 168.36° tilted at an angle of about 5°. The water droplets were continuously and uniformly dropped, simulating natural droplets impacting the surface, starting from an ambient temperature of 13.7 °C and a relative humidity of 79.3%, up to a temperature of −25 °C. The water droplets accumulated and froze on the untreated surface, forming an ice layer. On the contrary, when impacting on the laser treated SHS, the droplets quickly bounced off and rolled off, thus determining an anti-icing behavior of the laser machined PTFE surfaces.

Laser texturing of silicone rubber was also exploited to obtain superhydrophobic performance. In [115], several micro-nano combined structures were directly prepared using a 200 ns laser at a wavelength of 1064 nm. A repetition rate of 100 kHz was set, and the fluence was changed from 7.5 to 15 J/cm^2^. Due to the ablation, a micro roughness was achieved. With an increase of fluence to 10 J cm^−2^, the laser-textured microstructures were covered with sharp peaks and nano mastoid structures (Figure 10a). Moreover, using a laser fluence of 10 J cm^−2^, larger nanoparticle size and a greater micro-nano abundance were noticed than on the surface ablated with higher fluences textured as previously reported in [116]. This higher roughness resulted in a smaller contact area between the substrate and the water droplet. Thus, more air was trapped at the interface, which further reduced the heat transfer efficiency between the droplets and the surface, thus allowing a delay in the freezing of a deposited droplet. Moreover, the air stored between the surfaces and the ice layer considerably reduced the ice adhesion strength of the textured surface compared to the pristine silicone rubber sample for a given ice layer area and thickness. The contact angle of the water droplets for all of the laser textured surfaces decreased with the temperature (Figure 10b), while the SA increased. This was due to the fine condensed water droplets generated on the sample surface during the cooling process, which had a considerable pulling effect on the dropped water droplet. Additionally, in this case, a laser fluence of 10 J cm^−2^ guaranteed the best hydrophobic performance during the cooling process of the silicone rubber substrate. At this fluence, before freezing, the contact angle of the droplet finally stabilized at about 140°.

Superhydrophobic surfaces with small sliding angles were also obtained by irradiating surfaces with fs-laser pulses at fluences close to the ablation threshold to produce Laser-Induced Periodic Surface Structures (LIPSS) [117]. LIPSS are believed to originate from the interference between the incident laser light and the scattered tangential wave originated from former laser pulses [29]. Usually, LIPSS consist of regular rippled structures with a periodicity close to the incident laser wavelength and orientation perpendicular to the polarization of the incident light [30]. Several materials, like, e.g., stainless steel [118] and Ti–6Al–4V [119], have been found to become superhydrophobic after structuring with LIPSS. The combination of fs-laser direct laser writing producing ablation grooves followed by a second step irradiation to generate LIPSS on top of the non-ablated areas (Figure 11) has also been demonstrated to generate SHSs [120].

Another approach to producing SHS with multi-scaled hierarchical surface structures consists of combining Direct Laser Writing and Direct Laser Interference Patterning (DLIP) [121]. During the last decade, the DLIP technique has attracted growing interest due to its efficacy for producing highly controllable, flexible, and reproducible texture patterns from the micro to the sub-micro scale [122]. This technique consists of the coherent overlap of two or more laser beams. Modulating the laser intensity incident to the sample surface, an interference pattern is obtained. Different periodical features can be machined by controlling the angle between the interfering beams and changing the number of incident beams [123].

Milles et al. [124] studied the ice-repellent properties of pure aluminum, comparing three different surface arrangements: (i) triangular-like structures sequentially formed by three separate DLW rotated by 60°, each one producing line-like structures 50 μm apart; (ii) pillar-like structures fabricated using DLIP [125]; and (iii) multiscale textures due to the combination of DLW and DLIP, as illustrated in Figure 12a. Once in a stable wetting condition (after an aging time of 36 days in ambient air), the CA, CAH and SA were measured. The CA of the untreated reference surface had an average value of 93°, meanwhile all the three structured samples presented a SH behavior determined by the air cushions created between the water droplet and the structured surface leading to the Cassie–Baxter state. Regarding the dynamic wettability study, in the case of the DLW machined surface, the contact angle hysteresis (55°) and the sliding angle (34°) were drastically higher than the values measured for the DLIP (31° and 11°, respectively) and DLW + DLIP (36° and 13°, respectively) treated samples. These discrepancies were ascribed to a Wenzel regime, probably established during wetting of the DLW samples, while for both the DLIP and DLW + DLIP samples, the Cassie–Baxter wetting state prevailed. The authors did not provide an explanation to this difference, which definitely requires further investigation. The CA of 8 µL droplets placed on the samples held at temperatures ranging from −30 °C to 80 °C was studied to evaluate the influence of the temperature on the wetting behavior. As emerged from Figure 12b, three different regimes can be identified. In regime A, namely with temperature ranging from 20 °C to 80 °C, the CA of all the surfaces did not show any substantial difference with the temperature. On the contrary, decreasing the temperature down to −20°C (regime B), the authors observed a partial spreading of the droplet, with a linear decrease of the CA with the temperature for all the examined samples. Such behavior was attributed to the condensation of moisture from the environment on the materials [126] and to the following frosting front generated by ice nucleation from the vapor phase as the temperature further decreased, thus leading to the droplet spreading [56]. For temperatures below −20 °C (regime C), the droplet began to freeze at its first contact with the surface. Thus, the droplets did not fully spread, generating an increase in CA when decreasing the temperature. 

To characterize the anti-icing properties of the laser-fabricated surfaces, the freezing time of 8 µL water droplets put onto the surface was measured at −20 °C, as shown in Figure 12c. The droplets deposited on the treated surfaces froze, on average, after a time three times longer than the untreated reference. These results were promising for a significant delay in ice formation in real application environments, too. Interestingly, plotting the freezing time as a function of CA for every single measured droplet, a positive correlation was found, regardless of the surface topography. From the heat transfer simulations based on the finite element method, it emerged that the reason for the freezing delay observed could be attributed to the larger contact angles of the machined samples, which result in a smaller contact area between the droplets and the surface. Conversely, from the simulations, the increase in the thermal resistance due to the air trapped in the structures could explain an increase of the freezing time of only 6%. Therefore, this was not the main reason for the delayed ice formation.

A set of design rules for generating superhydrophobic Ti6Al4V (Ti64) surfaces with icephobic properties was defined in [127]. This very recent and exhaustive experimental work aims to investigate the adhesion strength of impact ice (i.e., ice generated close to operation conditions in icing wind tunnel tests) in order to prevent ice growth on external aircraft surfaces. DLW, LIPSS, and DLIP fabrication techniques, using laser sources with pulse durations ranging from the nanosecond to the femtosecond range, were employed. The authors found that all the used combinations of laser parameters and techniques conferred superhydrophobic properties to Ti64 samples, with CA higher than 150° and sliding angles lower than 15°, which are typical values for a Cassie–Baxter wetting state. Ice adhesion tests were performed in a icing wind tunnel iCORE (icing and COntamination REsearch facility [128]). This facility made it possible to reproduce atmospheric in-flight icing conditions generating a cloud of typical supercooled water droplets with a median volume diameter of ≈20 μm [128]. Four typical atmospheric ice types were reproduced changing the freezing fractions (FF) [129]. Once the ice was formed on the sample surfaces, the interfacial shear stress between the ice and the surface was measured [80]. Regardless of the atmospheric icing conditions (i.e., FF) the lowest values of the ice adhesion strength were obtained for the DLIP and LIPSS microstructures. The most plausible explanation of such results relies on the size of the surface structures that in case of DLIP or LIPSS is so small to repel the micrometric supercooled water droplets, thus preserving the superhydrophobicity of the surfaces (see Figure 13a). In this way, the Cassie–Baxter wetting state is preserved, inducing a lower ice adhesion compared to the pristine reference surface of Ti64. Conversely, surface structures with a size or spatial period comparable to the droplet dimension are not capable of efficiently repelling water and ice, since the supercooled water droplets are small enough to fill up the air gaps between the features (Figure 13b). In this case, a larger contact area produces a mechanical interlocking between ice and surface structures with a consequent stronger adhesion compared to the reference surface. In conclusion, the authors ascribed the improved anti-icing performances to two mechanisms: (i) being in a Cassie state, the contact area between the accreted ice and the surface is lower than on a flat surface; and (ii) the air pockets left between the surface and the ice act as stress concentrators. 

Recently, for the first time, Alamri et al. [14] reported on the de-icing behavior of the Direct Laser Interference Patterning of curved aerodynamic profiles, namely airfoils, combined with electrothermal heating used for ice protection systems. After the laser treatment, the airfoil was treated with MecaSurf, a commercial product used to reduce the surface free energy to make it strongly superhydrophobic, with a WCA exceeding 150°, and a roll-off angle < 15. When microstructured Ti6Al4V airfoils were placed in an icing wind tunnel and exposed to different icing conditions, DLIP was demonstrated to reduce the power needed to keep an airfoil free of ice by up to 80% while in controlled icing conditions. Under atmospheric impact icing conditions, at high air speeds, it was also demonstrated that aerodynamic forces alone were sufficient for passive anti-icing of the laser-structured surfaces.

A final consideration must be taken regarding the scalability of laser processes. None of the above-mentioned laser techniques, from DLW to LIPSS, despite their promising results in anti-icing, appear suitable for large-scale applications. Nevertheless, many large-scale approaches could be involved to obtain a throughput compatible with industrial environment. The on-the-fly method for synchronizing a laser galvanometer scanner and a linear stage for quick and wide area fabrication could be promising in this direction [130]. Moreover, the roll-to-roll technique [131] could be a good candidate, too, for helping laser processing to become an enabling technology for the manufacture of anti-icing surfaces at an industrial scale. However, no studies on large-scale laser texturing for anti-icing could be found in the literature. This could be a useful direction of future research investigation.

## 5. Conclusions

Undesired ice formation poses an increasingly serious threat in many industrial fields, particularly in aeronautics and telecommunications installations. Currently, the main strategies for reducing the problem are still based on active traditional methods such as mechanical, electro-thermal and liquid hybrid approaches. These approaches are definitively inefficient, energy-consuming, expensive, and can cause systems failure. Passive anti-icing coatings have also been proposed as an alternative to reduce or even replace the traditional de-icing methods. However, the existing available techniques present some disadvantages, such as their short-lived effectiveness, high manufacturing costs, and, in the case of chemical methods, their high environmental impact.

In recent years, several studies have demonstrated a correlation between superhydrophobicity and anti-icing properties. Based on the surface wetting theory, substrates characterized by surface micro–nanostructures and low surface free energy can well conform to the wetting regime described by the Cassie–Baxter model. Such surfaces exhibit high contact angles (CA), low hysteresis (CHA) and sliding angles (SA) and, consequently, low droplet adhesion. In sub-zero temperature environments, high CAs usually result in an increase of the droplet freezing time, while low CHAs and SAs lead to a short contact time between dripping droplets and the surfaces, thus hindering ice formation. Indeed, low ice adhesion has been reported on SHSs.

Among all the technologies for changing surface topography and obtaining SHSs, laser micromachining is considered to have many advantages. It offers extreme flexibility with respect to the morphology of the structures and surface micro-/nano-features that can be created, without posing any restrictions as to the type of material to be treated. The possibility of scaling up such technology to structure large areas with a reduction in process time and cost is also very attractive for industrial implementation.

Several approaches (i.e., DLW, to LIPSS and DLIP) in different ranges, from nano- to microscale, have been reported in the literature to produce SHSs with anti-icing properties. In all cases, a strong influence of the machined structures on the anti-icing behavior was found. Regardless of the materials, a texture smaller than the droplet size may result in an anti-icing behavior with a delay in the droplet freezing time. This can be explained with both ice nucleation theory and thermodynamics. In fact, the nucleation rate decreases when CA increases, delating the ice formation. Moreover, the air trapped between the water droplet and the rough surface, acting as a thermal insulator, contributes to a reduction in the heat transfer between the solid and the liquid surfaces. Similarly, it was demonstrated that the water repellence can induce supercooled water droplets dripped onto the SH material to bounce away rapidly from the surface before completely freezing. Finally, even though an ice film is eventually formed on the functionalized surface, the superhydrophobic wetting regime may induce a smaller contact interface of solid/ice, thus allowing a lower ice adhesion and the pursuit of the aim of preventing ice accumulation.

## Figures and Tables

**Figure 1 materials-13-05692-f001:**
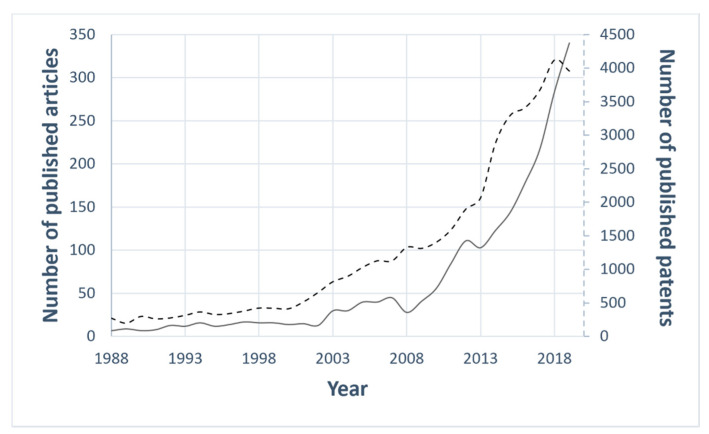
The statistics of the published articles indexed in Scopus (black line) and of the published patents listed in Espacenet [15] (black dotted line) on the topic of “anti-icing”. Year range selected: 1988–2019.

**Figure 2 materials-13-05692-f002:**
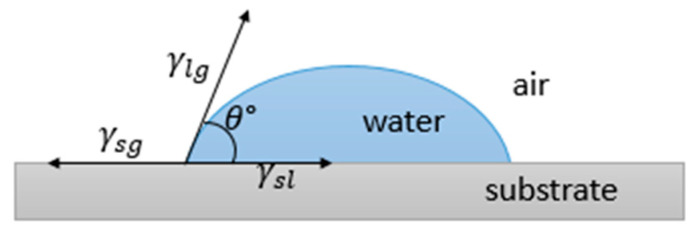
Contact angle of a liquid lying on a surface and balance between surface tensions.

**Figure 3 materials-13-05692-f003:**
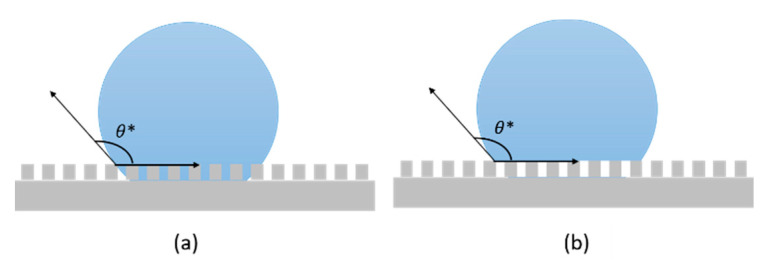
Schematic of droplet in (**a**) Wenzel state and (**b**) Cassie and Baxter state.

**Figure 4 materials-13-05692-f004:**
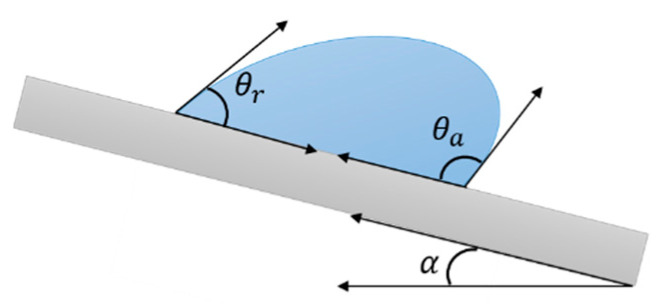
Schematic of advancing and receding angles of a droplet sliding on a surface. Contact angle hysteresis is defined as the difference between these two.

**Figure 5 materials-13-05692-f005:**
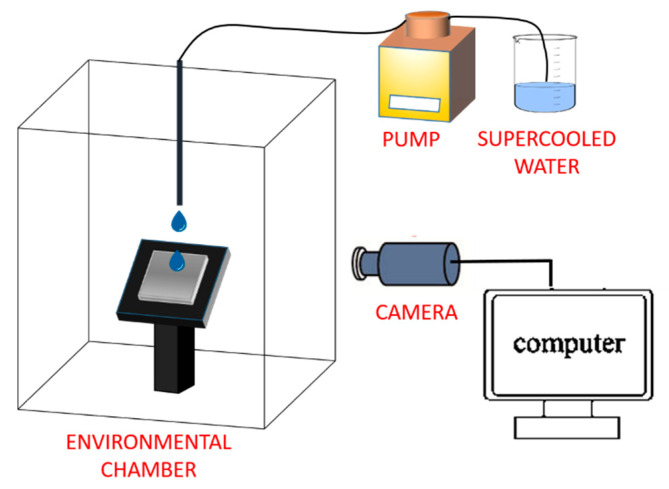
Schematic of a typical experimental setup for water dripping tests.

**Figure 6 materials-13-05692-f006:**
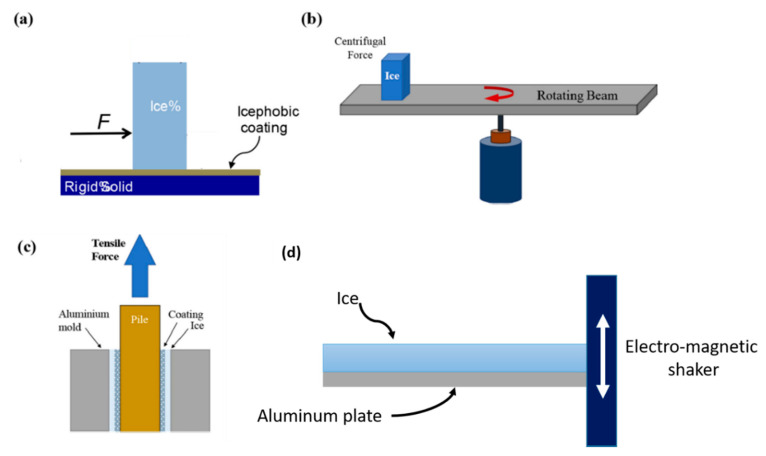
Schematic of the most common experimental methods for evaluating ice adhesion. (**a**) Peak force method. (**b**) Centrifugal force method. (**c**) Tensile force method. (Reprinted from [79], with permission from Elsevier). (**d**) Lateral view of the clamped ice-aluminum beam employed in the dynamic vibration test [80].

**Figure 7 materials-13-05692-f007:**
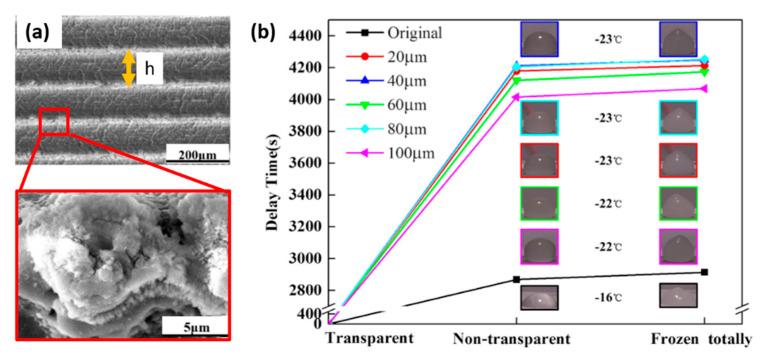
(**a**) SEM image of the laser processed surface by parallel scanning lines. Highlighted in red is an enlarged image of the cauliflower-like protrusions spread on the micro-gratings. (**b**) Delay times as a function of the pitch distance h (Adapted from [65], with permission from Elsevier).

**Figure 8 materials-13-05692-f008:**
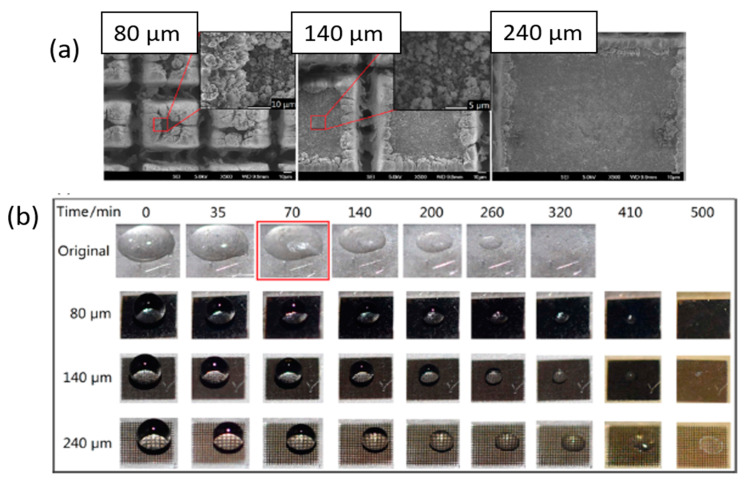
(**a**) Scanning electron microscope images of the stainless steel laser textured samples. Groove pitches: 80, 140, and 240 μm. In the insets, magnifications of the redeposited aggregates. (**b**) Evolution of the droplet set on the original and laser-machined superhydrophobic surfaces at subzero temperature (−8.5 ± 0.5 °C) (Reprinted with permission from [109]).

**Figure 9 materials-13-05692-f009:**
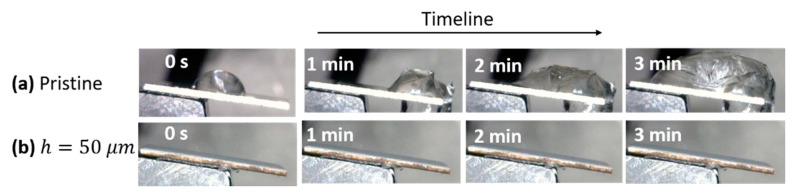
Snapshots of water accumulation (**a**) on the pristine surface compared with (**b**) the laser-treated Al2024 one at −20 °C. (Reproduced from [111] under the Creative Commons Attribution License).

**Figure 10 materials-13-05692-f010:**
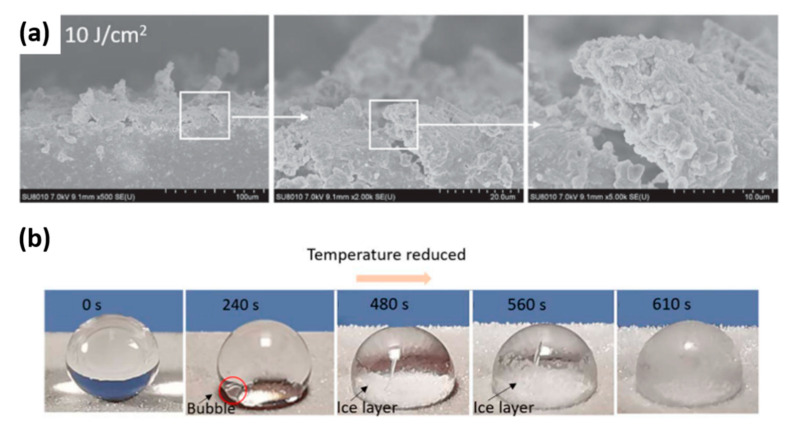
(**a**) Morphology of samples irradiated with laser fluences of 10 J cm^−2^. (**b**) Freezing of the water droplet set on the superhydrophobic silicone rubber surface (laser fluences 12.5 J cm^−2^). In the red circle a bubble squeezed into the water droplet during the process is highlithed, Adapted from [115] (© IOP Publishing. Reproduced with permission. All rights reserved).

**Figure 11 materials-13-05692-f011:**
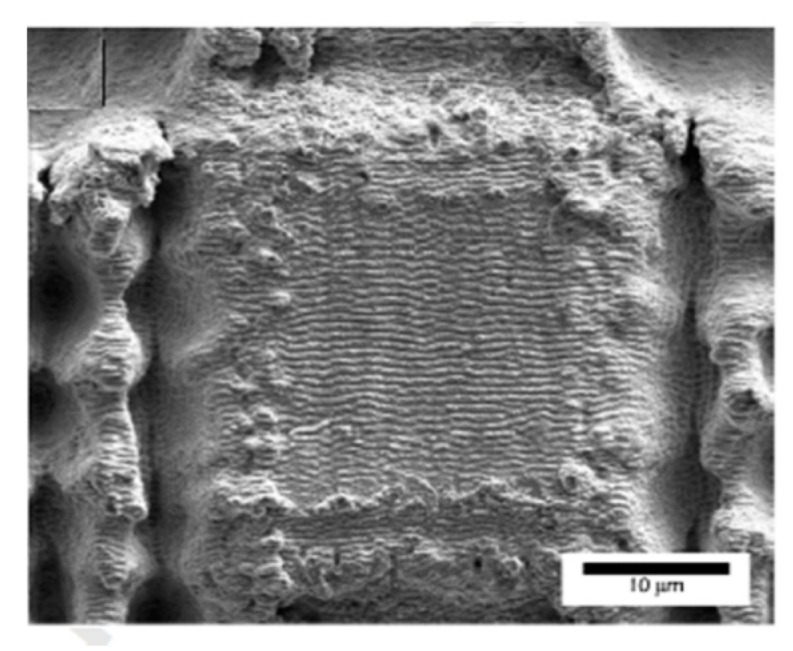
SEM image of laser fabricated hierarchical structures: matrix micropattern with a pitch distance of 50 μm covered by LIPSS (Reprinted from [120], with permission from Elsevier).

**Figure 12 materials-13-05692-f012:**
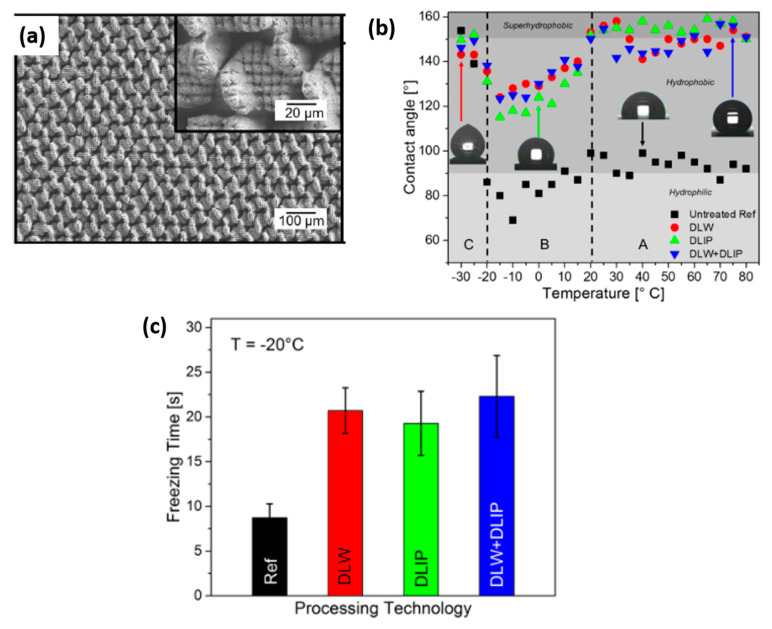
(**a**) Hierarchical microstructures, via DLW and DLIP techniques on pure aluminum. (**b**) Comparison between the contact angle of untreated reference, DLW, DLIP, and DLW + DLIP textured samples. The inserted images represent the droplets providing the measured angles depending on the temperature. (**c**) Average freezing time on an untreated reference, DLW, DLIP, and DLW + DLIP structures at −20 °C. The experiments were performed using 8 µL of deionized water (From [124] under http://creativecommons.org/licenses/by/4.0/).

**Figure 13 materials-13-05692-f013:**
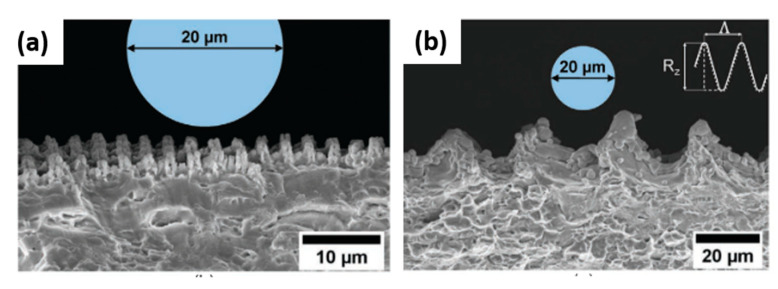
Side view SEM image of two laser textured surfaces with a spatial period (**a**) bigger than and (**b**) comparable to a water droplet of 20 µm. (Reproduced from [124] under http://creativecommons.org/licenses/by/4.0/).

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
