# Peer review of "Laser Fabrication of Anti-Icing Surfaces: A Review"

_materials, 2020, doi:10.3390/ma13245692_

Round 1

Reviewer 1 Report

This paper presents a review of the current use of lasers to alter the surface characteristics which are prone to ice accumulation. There are number of industries citied which are adversely affected by ice accretion. In particular ice accretion on general aerodynamic surface may represent serious safety implications.

It would be useful at an early stage in the article to distinguish between anti-icing and de-icing. Line 37 of the text focuses of anti-icing and draws reference to the accumulation of ice requiring removal. Ice protection systems (IPS) can be classified under two groups: anti-icing, which is aimed at preventing ice formation whereas de-icing is aimed at removing already established ice accumulation.

De-icing fluids are used for commercial aircraft in the northern hemisphere and comprise various fluids. They are essential not just on the exposed surfaces of aircraft wings but used on the aerodynamically quiet areas such as under wing and rear spar stabilizer areas. Two tests not mentioned are the Wet Spray Endurance Test and Boundary layer displacement thickness which is a measure of the remaining film thickness on a surface during its continuous removal by an increasing flow of air over it. Both WSET and BLDT are critical to safe flight operation and are closed associated with density, surface tension and viscosity of fluids. Testing is undertaken in climatically controlled wind tunnels. Some useful references are:

Kotker, D Deicing/Anti-icing fluids in Aero 1999, Boeing Commercial Airplanes Group, Seattle, WA USA

Cal A, Training recommendations and background information for de-icing/anti-icing of aircraft on the ground, 2008

Hille J De-icing and anti-icing fluid residues in Aero2007, Boeing Commercial Airplanes Group WA Seattle pp14-21

Some minor points:

L24 aircraft

L34 Remove “Despite”

L46 Salt depresses the freezing point down to the eutectic point. It does not melt ice per se.

L51 Useful to have a reference here

L146 English. Replace “than the Wenzel’s one” with “than that by Wenzel.”

L396 metal surfaces

Reviewer 2 Report

In her manuscript, Dr. Volpe shows the recent achievements on the laser engraving of metallic and polymeric surface for achieving icehobic surfaces. The review work is valuable, presents a comprehensive theorethical part and reports recent works on laser micromachining for anti-icing surfaces in a correct and clear manner. Neverthelss some inaccuracies regarding the presentation of the results can be further corrected and the overall form of the paper can be improved.
Therefore I suggest accepting the submission after minor revisions. In the following, some aspects that should be addressed in the second revision phase.

1) In the introductory part, the authors spent very few words in describing the problems related with icing and ice formation.
Moreover, in view of the following sections, the authors neglected the description of the differences between icing on ground (e.g. snow accumulation, frost, etc.) and icing on board (on aircrafts or rotorcrafts).
Which of the techniques mentioned in Section 3 are suitable to test the first type of ice? Are there techniques useful to test the second type?

2) The formatting of Figure 1 can be improved. I recommend to remove the title from the graph and change "documents" with "number of published articles". Moreover, I suggest to include on this plot the number of published patents in a second Y-axis, by means of a search in pat-base or other patent search motors.

3) Line 376: "DLIP" instead of "DLIPS".

4) It is highly recommended to enlarge Figure 8b. The images are barely visible at 100% magnification.

5) The authors mention Reference 110 as a method for producing SHSs. Although the authors previsouly mention that a correlation between SHSs and ice-repulsion is given, no indication is provided that the microstructures mentioned in the paper are also ice-repellent.
As a general advice, since the authors called Section 4 "Fabrication of superhydrophobic surfaces with anti-icing properties" I recommend to discuss only works that strictly deal with the fabrication of icephobic textures.

6) The authors mention several times the DLIP as an effective method for producing surfaces with a sensible anti-igin behaviour.
A more recent article regarding this approach, showing further icing mechanisms has been published here: https://onlinelibrary.wiley.com/doi/full/10.1002/admi.202001231

Reviewer 3 Report

This article reviews the most relevant works on the use of laser technology to functionalize surfaces with anti-icing properties.  As the authors point out, there is an increasing interest for anti-icing surfaces in the last 10 years but also several challenges to be solved. The article is well structured, but linguistic revision is needed.  Some sentences are unclear.  The article should be published after minor revision.  For article publication, equation (9) must be clarified. Specifically:

Line 9: “in many fields” at the end of the sentence should be removed as it already at the start of the sentence. 

Line 13:  “have been discussed” should be replaced by “is discussed”.

Line 23: forms appears two times in the same sentence.  It should be corrected.

Line 76: a word is missing “..the surfaces to exhibit not only of the high water contact”

Line 112: not clear “to a distance of infinity”

Line 136, equation 3, define actual surface and geometrical surface

Line 242, equation 8, what about the latent heat of solidification?

Line 248, equation 9: there is a problem with the units of this equation.  From what I understand: rho(kg/m3), cp(J/kg K), T(K).  What is the unit of delta h, knowing  that Tf should be Kelvin.  Also, as there is no time in this equation, at line 256, you cannot directly conclude “causing slower freezing”.   

Line 284: a word or punctuation is missing “by a pump An optical”

Line 321: in the closure paragraph of section 3, a link with the section 4 is missing.

Line 345: correction needed “heat loss through cheat conductions”

Line 351: remove a word “A reduction up to the 50% of the ice…”

Line 400: missing words? “… properties of the as laser-treated materials”

Lines 426-429: this is not clear for me.  If the heat transfer to the air increase, is it heat loss or heat gain?

Line 513: should be figure (b) not (c).

Line 592: define impact ice or remove.  This is the first use of impact ice in the manuscript.

Line 622-623: the sentence starting with “Nevertheless…” is not clear.

Line 634-635: What is the meaning of “serious damage to fatigue properties of the substrate materials”

Line 637: clarify “their short duration”

Line 657: unclear “…being the solid-liquid free-energy barrier dependent on it”

Reviewer 4 Report

This article is a good review paper on icing phenomena and making anti-icing surfaces with laser. Since it covers a wide range of technical papers published in the literatures, it would greatly help readers to understand icing and laser fabrication of anti-icing surfaces. I think this article is suitable for publication in the journal "Materials" after minor revision listed below.

(1)The equation numbers are incorrect from Page 4. Please correct them and the corresponding sentences.

(2)The sub-numbering of Fig.10(b) in Page 13 are confusing, because (a), (b) are duplicated. Please use different sub-numbering.

That's all.     

Round 2

Reviewer 2 Report

Dr. Volpe and co-authors improved the manuscript according to the changes suggested in the previous revision phase. Therefore, I reccomend to accept the manuscript in the present form, with no further changes.

Reviewer 4 Report

Thank you for properly revising the paper.